# Covid-19 in end-stage renal disease patients with renal replacement therapies: A systematic review and meta-analysis

Tanawin Nopsopon[1], Jathurong Kittrakulrat[2], Kullaya Takkavatakarn[3], Thanee Eiamsitrakoon[4], Talerngsak Kanjanabuch[3], Krit Pongpirul[1,5,6]*

1 Department of Preventive and Social Medicine, Faculty of Medicine, Chulalongkorn University, Bangkok, Thailand, 2 Department of Medicine, Faculty of Medicine, Chulalongkorn University, Bangkok, Thailand, 3 Division of Nephrology, Department of Medicine, Faculty of Medicine, Chulalongkorn University, Bangkok, Thailand, 4 Division of Nephrology, Department of Medicine, Faculty of Medicine, Thammasat University, Pathum Thani, Thailand, 5 Department of International Health, Johns Hopkins Bloomberg School of Public Health, Baltimore, Maryland, United States of America, 6 Bumrungrad International Hospital, Bangkok, Thailand

* doctorkrit@gmail.com

**Data Availability Statement:** All relevant data are within the manuscript and its Supporting Information files.

## Abstract

### Background

The novel coronavirus (COVID-19), caused by SARS-CoV-2, showed various prevalence and case-fatality rates (CFR) among patients with different pre-existing chronic conditions. End-stage renal disease (ESRD) patients with renal replacement therapy (RRT) might have a higher prevalence and CFR due to reduced immune function from uremia and kidney tropism of SARS-CoV-2, but there was a lack of systematic study on the infection and mortality of the SARS-CoV-2 infection in ESRD patients with various RRT.

### Methodology/Principal findings

We searched five electronic databases and performed a systematic review and meta-analysis up to June 30, 2020, to evaluate the prevalence and case fatality rate (CFR) of the COVID-19 infection among ESRD patients with RRT. The global COVID-19 data were retrieved from the international database on June 30, 2020, for estimating the prevalence and CFR of the general population as referencing points. Of 3,272 potential studies, 34 were eligible studies consisted of 1,944 COVID-19 confirmed cases in 21,873 ESRD patients with RRT from 12 countries in four WHO regions. The overall pooled prevalence in ESRD patients with RRT was 3.10% [95% confidence interval (CI) 1.25–5.72] which was higher than referencing 0.14% global average prevalence. The overall estimated CFR of COVID-19 in ESRD patients with RRT was 18.06% (95% CI 14.09–22.32) which was higher than the global average at 4.98%.

### Conclusions

This meta-analysis suggested high COVID-19 prevalence and CFR in ESRD patients with RRT. ESRD patients with RRT should have their specific protocol of COVID-19 prevention and treatment to mitigate excess cases and deaths.

**Funding:** The authors received no specific funding for this work.

**Competing interests:** The authors have declared that no competing interests exist.

## Author summary

Chronic kidney disease (CKD) was associated with increasing severity and mortality of COVID-19. End-stage renal disease (ESRD) patients were at the terminal stage of CKD and had reduced immune function due to uremia. Additionally, ESRD patients with kidney transplantation had a diminished immune system from immunosuppressive agents. Kidneys might be the secondary target of SARS-CoV-2 after the respiratory tract regardless of the previous history of kidney disease, preferably the glomerulus, which was associated with the richness of some specific protein-coding genes in the kidney. The overall pooled prevalence in ESRD patients with renal replacement therapy was approximately 22 times the referencing global average prevalence. The overall estimated case fatality rate of COVID-19 in ESRD patients with renal replacement therapy was approximately 3.6 times the global average. ESRD patients with renal replacement therapy had high COVID-19 prevalence and case fatality rate. We suggested that ESRD patients with renal replacement therapy should have their specific protocol of COVID-19 prevention and treatment to mitigate excess cases and deaths.

## Introduction

The Severe Acute Respiratory Syndrome Coronavirus 2 (SARS-CoV-2) was first reported in December 2019 as a newly discovered causative pathogen of patients with pneumonia of unknown cause in Wuhan, China [1]. Since its debut, the novel coronavirus (COVID-19) has drastically spread throughout the globe, and the World Health Organization (WHO) declared COVID-19 as a pandemic on March 11, 2020 [2]. The global COVID-19 data from the international database at mid-year of 2020 reported 217 countries affected by the novel coronavirus with more than ten million people infected with SARS-CoV-2; approximately half a million patients died from the novel disease [3].

Pre-existing comorbidities have been reported to be associated with the severity of hospitalized COVID-19 patients [4]. Chronic kidney disease was found to be associated with the severity and mortality of COVID-19 [5,6]. Specifically, patients with chronic kidney disease had a three-fold risk of developing severe COVID-19 [7]. About one-third of end-stage renal disease (ESRD) patients with dialysis who were hospitalized with COVID-19 were died [8]. The high mortality rate might be associated with the diminished immune system from uremia in ESRD patients [9]. Moreover, kidney transplanted patients had reduced immune response from immunosuppressive agents [10].

Kidneys might be the secondary target of SARS-CoV-2 after the respiratory tract regardless of the previous history of kidney disease as indicated from the second-highest SARS-CoV-2 viral load found in the kidneys of the autopsy in COVID-19 death cases, preferably the glomerulus, which was associated with the richness of protein-coding genes including angiotensin-converting enzyme 2 (*ACE2*), transmembrane serine protease 2 (*TMPRSS2*), and cathepsin L (*CTSL*) in the kidney [11]. There was a high number of hospitalized COVID-19 patients who subsequently developed acute kidney injury (AKI) during the disease course [5,6], especially patients with kidney transplantation [12]. Several systematic reviews reported the association of AKI in COVID-19 patients with more severity and poor prognosis of COVID-19 [13–15]. Despite the evidence on the association of COVID-19 and subsequent kidney damage, whether patients with severe chronic kidney disease, especially those with renal replacement therapy (RRT), had higher COVID-19 infection and death than individuals with normal kidney

functions have still been inconclusive. In this review, we performed a meta-analysis to evaluate the prevalence and case fatality rate of COVID-19 in patients with ESRD with RRT.

## Methods

This study was conducted following the recommendations of the Preferred Reporting Items of Systematic Reviews and Meta-Analyses (PRISMA) statement [16]. We prospectively registered the systematic review with PROSPERO International Prospective Register of Ongoing Systematic Reviews (Registration number: CRD42020199752).

### Search strategy

PubMed, Embase, Scopus, Web of Science, and Cochrane Central Register of Clinical Trials were used to systematically search for articles published in the English language up to June 30, 2020. The terms "Coronavirus", "COVID-19" and "SARS-CoV-2" were used in combination with "Chronic kidney disease", "End-stage renal disease", "Renal replacement therapy", "Dialysis", "Hemodialysis", "Peritoneal Dialysis" and "Kidney Transplantation" as the keywords for literature search along with their synonyms. The search strategy is presented in detail in S2 Table. Additionally, the reference lists of included articles were searched, as well as related citations from other journals via Google Scholar.

### Study selection

We worked with an information specialist to design an appropriate search strategy to identify original peer-reviewed articles of randomized controlled trials and observational studies evaluating the prevalence or mortality outcomes, or both of COVID-19 in ESRD patients with RRT (KT, HD, or PD) without restriction on age, gender, ethnicity, duration of chronic kidney disease, or previous treatment. Additional outcomes were the need for mechanical ventilation (MV) and intensive care unit (ICU) admission. Article screening was done by two independent reviewers for eligible studies. Discrepancies between the two reviewers were resolved by consensus.

### Data extraction

Data extraction was done by two independent reviewers for published summary estimate data. Discrepancies between the two reviewers were resolved by consensus. We extracted the following data: (1) study characteristics (authors, year of publication, study type, journal name, contact information, country, and funding), (2) patients characteristics (sample size, age, gender, type of renal replacement therapy, COVID-19 diagnostic criteria, COVID-19 treatment), (3) outcomes (complete list of the names of all measured outcomes, unit of measurement, follow-up time point, missing data) as well as any other relevant information. All relevant text, tables, and figures were examined for data extraction. We contacted the authors of the study with incompletely reported data. If the study authors did not respond within 14 days, we conducted analyses using the available data.

The global COVID-19 data were retrieved from the international database [3] on June 30, 2020, for estimating the prevalence and CFR of the general population as referencing points. The global COVID-19 data in general population comprised of 7,525,172,273 population, 10,566,205 COVID-19 confirmed cases, and 526,163 COVID-19 death cases in 217 countries. The global average prevalence of COVID-19 in the general population was 0.14%, and the global average CFR of COVID-19 in the general population was 4.98%.

## Quality assessment

The authors worked independently to assess the risk of bias in the included studies using the risk of bias tool by Hoy et al. for studies with COVID-19 prevalence outcomes [17]. We assessed the representativeness of the sample, sampling frame, sampling techniques, response rate, data collection method, case definition, measurement tools, study period, and data calculation. We assigned each domain as a low risk of bias and a high risk of bias while the overall risk of bias was reported as a low risk of bias, a moderate risk of bias, and a high risk of bias. The included studies with COVID-19 mortality data were assessed for risk of bias with the non-summative four-domain system (consecutive cases, multicenter, more than 80% follow-up, multivariable analysis) developed by Wylde et al [18]. We assigned each domain as adequate, inadequate, and not reported. This system was preferred because the domain was applicable with both controlled observational studies and case series. We resolve the disagreement through discussion. We presented our risk of bias assessment in S1 Table.

## Statistical analysis

The primary outcomes were COVID-19 prevalence and COVID-19 deaths. The prevalence outcomes were measured with the percentage of COVID-19 confirmed cases and total ESRD with RRT patients with an associated 95% confidence interval (CI). The death outcomes were measured as CFR with the percentage of COVID-19 deaths and total COVID-19 confirmed cases in ESRD with RRT patients with an associated 95% CI. The additional outcomes were the need for mechanical ventilation and ICU admission among hospitalized COVID-19 confirmed cases in ESRD with RRT patients with an associated 95% CI. The results of the studies were included in the meta-analysis and presented in a forest plot, which also showed statistical powers, confidence intervals, and heterogeneity. This meta-analysis was performed to find pooled estimated value with a 95% confidence interval and did not have a comparator group, thus no null hypothesis was tested. Instead, we compared the estimated values of primary outcomes with global averages.

We assessed clinical and methodological heterogeneity by examining participant characteristics, follow-up period, outcomes, and study designs. We then assessed statistical heterogeneity using the $I^2$ statistic for magnitude, direction, and strength of evidence for heterogeneity. We regarded level of heterogeneity for $I^2$ statistic as defined in chapter 9 of the Cochrane Handbook for Systematic Reviews of Interventions: 0–40% might not be important; 30–60% may represent moderate heterogeneity; 50–90% may represent substantial heterogeneity; 75–100% considerable heterogeneity. The Freeman-Tukey double arcsine transformation was used to ensure admissible confidence intervals. The random-effects meta-analysis by DerSimonian and Laird method was used as clinical, methodological, and statistical heterogeneity encountered. The exact method was used for confidence interval computation. Prespecified subgroup analyses, including country income level, WHO country region, and RRT modality, were performed. We assessed publication bias in two ways. First, we computed each study effect size against standard error and plotted it as a funnel plot to assess asymmetry visually. Second, we used Egger's test to statistically test for asymmetry. The significant asymmetry indicated the possibility of publication bias or heterogeneity. The meta-analysis was performed using STATA 16.1 (StataCorp, TX, USA).

# Results

## Characteristics of the studies

The database search identified 3,272 potential records. After duplicate removal, 1,885 titles passed the initial screening, and 212 theme-related abstracts were selected for further full-text

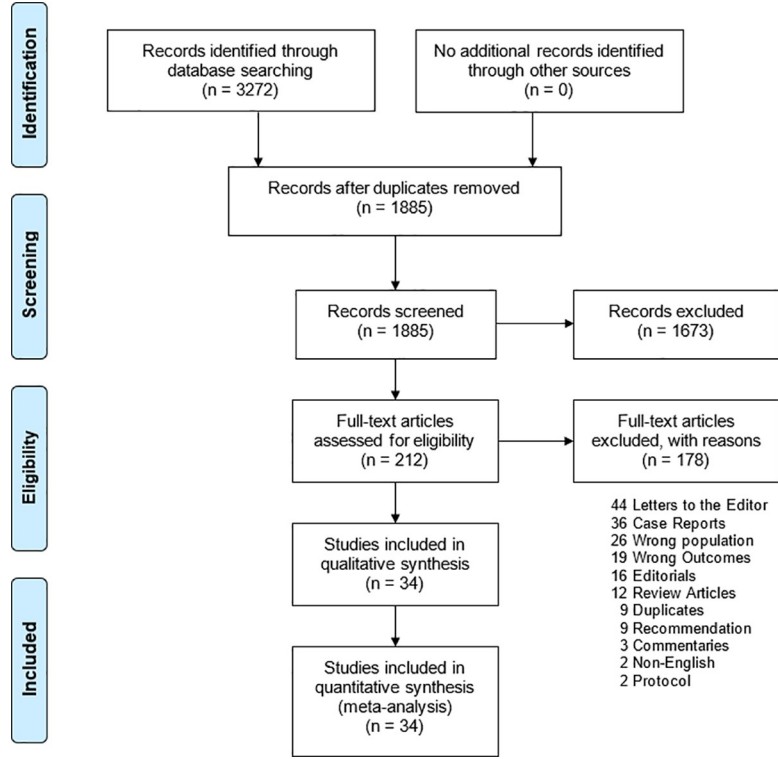

**Fig 1. Preferred Reporting Items for Systematic Reviews and Meta-Analyses flowchart of study selection and database search.**

articles assessment for eligibility (Fig 1). A total of 178 articles were excluded as the following: 44 letters to the editor, 36 case reports, 26 wrong populations, 19 wrong outcomes, 16 editorials, 12 review articles, nine duplicates, nine recommendation, three commentaries, two non-English, and two protocols. Only 34 studies were eligible for data synthesis and meta-analysis.

All 34 included studies were published in 2020, of which 15 studies reported prevalence outcomes [19–33] and 31 studies reported death outcomes [8,19,22–25,27–51]. There were 17 cohort studies [8,20,21,24–26,28,32,34,36,41,43,45–47,50,51], 12 case series [22,31,35,37–40,42,44,48,49], and five cross-sectional studies [23,27,29,30,33]. The number of samples per study ranged from 3 to 7,154 with a total of 1,944 COVID-19 confirmed cases in 21,873 ESRD patients with RRT from 12 countries in four WHO regions which nine countries were high-income countries and three were upper-middle-income countries according to the World Bank classification [52]. The mean age varied from 45.0 to 73.6 years. Female patients in each study were ranged from 0 to 57%. There were 20 studies on ESRD patients with kidney transplantation (KT) only [22,24,25,27,29,31,35–40,42,44,46–51], ten studies with hemodialysis (HD) only [19–21,23,26,28,30,33,34,41], four studies with more than one RRT modalities [8,32,43,45], and no studies with peritoneal dialysis (PD) only (Table 1).

Twenty-seven studies used only the polymerase chain reaction (PCR) assay as the diagnostic criteria to confirm the COVID-19 case. Five studies used a combination of PCR and additional criteria to diagnose COVID-19: imaging (three studies), suggestive symptoms (one study), and a combination of imaging, suggestive symptoms, and laboratory tests (one study). One study did not explicitly state the diagnostic criteria. For COVID-19 treatment, Hydroxychloroquine was the most common treatment used in 20 studies, followed by Lopinavir/Ritonavir used in 12 studies. Some studies used other medications including Darunavir/Ritonavir,

**Table 1. Characteristics of included studies.**

| Author | Country | Income Level | WHO Regions | Study Design | Mean Age, yr (SD) | Female % | Total Sample | COVID-19 Cases | RRT Modality | Diagnostic Criteria | COVID-19 Treatment | Admission Cases | IV Steroid Cases |
|---|---|---|---|---|---|---|---|---|---|---|---|---|---|
| Wang[19] | China | UMICs | Western Pacific | Case Series | Range 47–67 | 40.0* | 201 | 5 | HD | PCR | Umifenovir, Ribavirin | 5 | NR |
| Wu[34] | China | UMICs | Western Pacific | Cohort | Med 62 (IQR 54–71) | 36.7 | 49 | 49 | HD | PCR | NR | 49 | NR |
| Xiong[20] | China | UMICs | Western Pacific | Cohort | 63.3 (13.2) | NR | 7154 | 154 | HD | PCR | NR | NR | NR |
| Xu[21] | China | UMICs | Western Pacific | Cohort | NR | 43.3 | 1542 | 5 | HD | PCR | NR | NR | NR |
| Zhang[22] | China | UMICs | Western Pacific | Case Series | 45 (11) | 20.0* | 743 | 5 | KT | NAT | Oseltamivir, Umifenovir | 5 | 1 |
| Zhu[35] | China | UMICs | Western Pacific | Case Series | Range 24–65 | 20.0 | 10 | 10 | KT | PCR, IMG | Umifenovir, Oseltamivir, Ribavirin, Ganciclovir | 10 | 5 |
| Bösch[36] | Germany | HICs | European | Cohort | Med 61 | 33.3 | 3 | 3 | KT | PCR | None | 3 | 0 |
| Abrishami[37] | Iran | UMICs | Eastern Mediterranean | Case Series | 47.66 (1.35) | 25.0 | 12 | 12 | KT | PCR, IMG, SYM, Lab | HCQ, LPV/r | 12 | 12 |
| Alberici a[23] | Italy | HICs | European | Cross-sect | Med 72 (IQR 62–79) | 34.0* | 643 | 94 | HD | PCR | HCQ, LPV/r, DRV/r | 57 | 18 |
| Alberici b[38] | Italy | HICs | European | Case Series | 59 | 20 | 20 | 20 | KT | PCR | HCQ, LPV/r, DRV/r | 20 | 11 |
| Maritati[24] | Italy | HICs | European | Cohort | Med 72 | 40* | 585 | 5 | KT | PCR | HCQ, LPV/r | 5 | NR |
| Mella[39] | Italy | HICs | European | Case Series | 55.5 (8.4) | 0 | 6 | 6 | KT | PCR, IMG | HCQ, DRV/r | 6 | NR |
| Hoek[25] | Netherlands | HICs | European | Cohort | Range 21–81 | 31.3* | 2150 | 16 | KT | PCR, SYM | NR | NR | NR |
| Kolonko[40] | Poland | HICs | European | Case Series | Med 42 | 0 | 3 | 3 | KT | PCR | None | 3 | NR |
| Cho[26] | South Korea | HICs | Western Pacific | Cohort | Med 57 | 36.4* | 1175 | 11 | HD | PCR | NR | 11 | NR |
| Jung[41] | South Korea | HICs | Western Pacific | Cohort | 63.5 (14.5) | 57.1 | 14 | 14 | HD | PCR | HCQ, LPV/r | 14 | 3 |
| Crespo[27] | Spain | HICs | European | Cross-sect | 73.6 (4.7) | 20* | 803 | 20 | KT | PCR | HCQ, LPV/r, DRV/r | 9 | NR |
| Fernández-Ruiz[42] | Spain | HICs | European | Case Series | Range 39–80 | 12.5 | 8 | 8 | KT | PCR | HCQ, LPV/r | 8 | NR |
| Goicoechea[28] | Spain | HICs | European | Cohort | 71 (12) | 36.1* | 282 | 36 | HD | PCR | HCQ, LPV/r | 36 | NR |
| Melgosa[43] | Spain | HICs | European | Cohort | NR | NR | 6 | 6 | KT, HD | PCR | HCQ, LPV/r | NR | NR |
| Rodríguez-Cubillo[44] | Spain | HICs | European | Case Series | Med 66 (IQR 59–72) | 41.4 | 29 | 29 | KT | PCR | HCQ | 29 | 18 |
| Sánchez-Álvarez[45] | Spain | HICs | European | Cohort | 68.3 (14.7) | NR | 868 | 868 | KT, HD, PD | PCR | HCQ, LPV/r | NR | NR |

*(Continued)*

Table 1. (Continued)

| Author | Country | Income Level | WHO Regions | Study Design | Mean Age, yr (SD) | Female % | Total Sample | COVID-19 Cases | RRT Modality | Diagnostic Criteria | COVID-19 Treatment | Admission Cases | IV Steroid Cases |
|---|---|---|---|---|---|---|---|---|---|---|---|---|---|
| Tschopp[46] | Switzerland | HICs | European | Cohort | Range 35–87 | 38.5 | 13 | 13 | KT | PCR | HCQ, LPV/r | 12 | NR |
| Akdur[29] | Turkey | UMICs | European | Cross-sect | NR | 31.8 | 509 | 1 | KT | PCR | HCQ | 0 | 0 |
| Arslan[30] | Turkey | UMICs | European | Cross-sect | 64 | 42.7 | 602 | 7 | HD | PCR, IMG | NR | NR | NR |
| Banerjee[31] | UK | HICs | European | Case Series | Med 54 | 42.9* | 2082 | 7 | KT | PCR | None | 5 | 0 |
| Corbett[32] | UK | HICs | European | Cohort | Med 66 (IQR 55–75) | 41.8 | 1530 | 300 | HD, PD | PCR | NR | NR | NR |
| Roper[33] | UK | HICs | European | Cross-sect | Median 63.2 | 40.4 | 670 | 76 | HD | PCR | NR | NR | NR |
| Chen[47] | USA | HICs | The Americas | Cohort | 56 (12) | 46.7 | 30 | 30 | KT | PCR | HCQ | 30 | NR |
| Columbia U[48] | USA | HICs | The Americas | Case Series | Med 51 (IQR 28–72) | 33.3 | 15 | 15 | KT | NR | HCQ | 15 | NR |
| Fung[49] | USA | HICs | The Americas | Case Series | Range 44–77 | 28.6 | 7 | 7 | KT | PCR | HCQ, LPV/r | 5 | NR |
| Mehta[50] | USA | HICs | The Americas | Cohort | Med 59 (IQR 53–64) | 34.3 | 35 | 35 | KT | PCR | HCQ | 34 | NR |
| Valeri[8] | USA | HICs | The Americas | Cohort | Med 63 (IQR 56–78) | 44.1 | 59 | 59 | HD, PD | PCR | HCQ | 59 | NR |
| Yi[51] | USA | HICs | The Americas | Cohort | NR | NR | 15 | 15 | KT | PCR | HCQ, Ribavirin | 11 | 1 |

*Female % of COVID-19 confirmed cases in end-stage renal disease patients with RRT.

Cross-sect, cross-sectional. DRV/r, Darunavir/Ritonavir. HCQ, Hydroxychloroquine. HD, hemodialysis. HICs, high-income countries. IMG, imaging. IQR, interquartile range. IV, intravenous. KT, kidney transplantation. LPV/r, Lopinavir/Ritonavir. Med, median. NAT, nucleic acid test. NR, not reported. PCR, polymerase chain reaction. PD, peritoneal dialysis. RRT, renal replacement therapy. SD, standard deviation. SYM, symptoms. UMICs, upper-middle-income countries. WHO, World Health Organization

Oseltamivir, Umifenovir, Ribavirin, and Ganciclovir. Three studies reported no specific COVID-19 treatment, and eight studies did not report information on COVID-19 treatment.

## Risk of bias

Of 15 included studies that reported prevalence, 12 had a mild overall risk of bias, and three studies had a moderate overall risk of bias. All of the included studies had a high risk of bias in the representativeness of the sample. Three studies had a high risk of bias in the sampling frame, sampling techniques, and response rate. Of 31 studies that reported mortality, 26 were not multicenter studies, 28 did not use multivariable analysis, 30 stated the inclusion of consecutive cases, and all studies had at least 80% follow-up (S1 Table). While the funnel plot showed some visual asymmetry, Egger's test for asymmetry highlighted no evidence of publication bias on both overall estimated prevalence (p = 0.418) and overall estimated case fatality rate (p = 0.569) (S1 Fig).

## Prevalence of COVID-19

Fifteen studies that reported the prevalence of COVID-19 in ESRD patients with RRT were included in the meta-analysis of 20,671 ESRD with RRT patients from seven countries [19–33]. The overall pooled prevalence was 3.10% [95% confidence interval (CI) 1.25–5.72] which was higher than referencing 0.14% global average prevalence (Fig 2). The highest COVID-19 prevalence in ESRD patients was reported in the United Kingdom [19.61 (95% CI 17.65–21.69)] [32] whereas the lowest COVID-19 prevalence was reported in Turkey [0.20% (95% CI 0.00–1.09%)] [29].

Six studies were from two upper-middle-income countries [19–22,29,30] and nine studies were from five high-income countries [23–28,31–33]. The pooled COVID-19 prevalence in

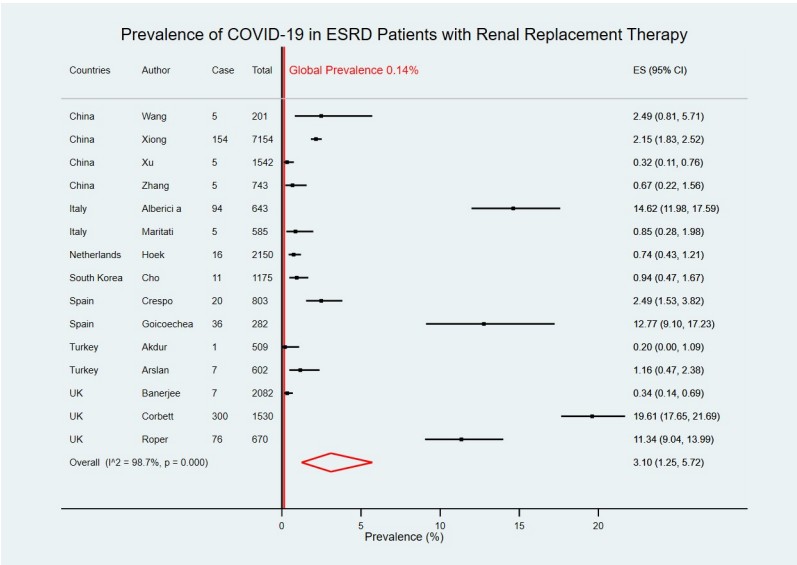

**Fig 2. Forest plot of the prevalence of COVID-19 in ESRD patients with RRT.** The figure summarizes the number of COVID-19 confirmed cases in end-stage renal disease (ESRD) patients with renal replacement therapy (RRT) and the total number of ESRD patients with RRT in 15 eligible studies. The forest plot represents the estimated prevalence of COVID-19 in ESRD patients with RRT for each study (black boxes), with 95% confidence intervals (95% CI; horizontal black lines). The estimated pooled prevalence (red diamond) was 3.10% (95% CI 1.25–5.72). The global average prevalence (vertical red line) was 0.14%. The meta-analysis used a random-effects model with the exact method for confidence interval estimation. ES, effect size. $I^2$, test for heterogeneity.

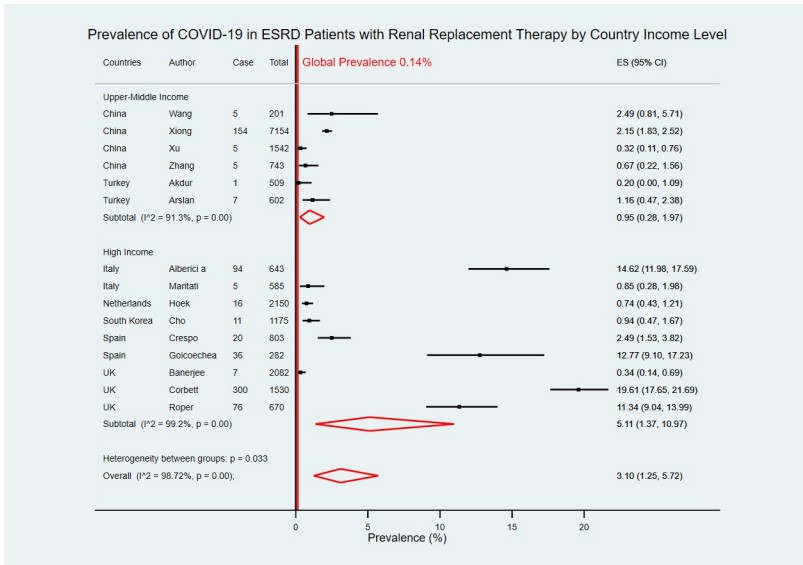

**Fig 3. Forest plot of the prevalence of COVID-19 in ESRD patients with RRT by country income level.** The figure summarizes the number of COVID-19 confirmed cases in ESRD patients with RRT and the total number of ESRD patients with RRT in 15 eligible studies with subgroup analysis by the World Bank country income level. The forest plot represents the estimated prevalence of COVID-19 in ESRD patients with RRT for each study (black boxes), with 95% confidence intervals (95% CI; horizontal black lines). The estimated pooled prevalence for each subgroup was presented with a red diamond. The overall estimated pooled prevalence (last red diamond) was 3.10% (95% CI 1.25–5.72). The global average prevalence (vertical red line) was 0.14%. The meta-analysis used a random-effects model with the exact method for confidence interval estimation. ES, effect size. $I^2$, test for heterogeneity.

upper-middle-income countries was 0.95% (95% CI 0.28–1.97) which was lower than the pooled prevalence in high-income countries at 5.11% (95% CI 1.37–10.97) (Fig 3). With WHO Region categorization, there were ten studies in the European region [23–25,27–33], and five studies in Western Pacific region [19–22,26]. COVID-19 estimated prevalence in the European region was 4.41% (95% CI 1.11–9.69) which was higher than the Western Pacific region at 1.10% (95% CI 0.36–2.19) (Fig 4).

Eight studies focused on COVID-19 prevalence in ESRD patients with HD only with a pooled COVID-19 prevalence of 4.26% (95% CI 1.68–7.91) [19–21,23,26,28,30,33]. Six studies had data on prevalence of COVID-19 in KT patients only with a pooled prevalence of 0.76% (95% CI 0.33–1.35) [22,24,25,27,29,31]. Only one study had mixed data of COVID-19 prevalence in ESRD patients with HD or PD with a prevalence of 19.61% (95% CI 17.65–21.69) (Fig 5) [32]. There was no study with ESRD patients with peritoneal dialysis only.

## Case fatality rate of COVID-19

Thirty-one studies with death outcomes among 1,774 COVID-19 confirmed cases in ESRD with RRT from 12 countries [8,19,22–25,27–51]. The overall estimated CFR of COVID-19 in ESRD patients with RRT was 18.06% (95% CI 14.09–22.32) which was higher than the global average at 4.98% (Fig 6).

Of 31 studies, seven studies were from three upper-middle-income countries [19,22,29,30,34,35,37] and 24 studies were from nine high income countries [8,23–25,27,28,31–33,36,38–51]. The estimated CFR of COVID-19 in ESRD patients with RRT in upper-middle-income countries was 8.95% (95% CI 0.00–30.00), while the estimated CFR in high-income countries was 19.65% (95% CI 15.94–23.60) (Fig 7). Nineteen studies were from

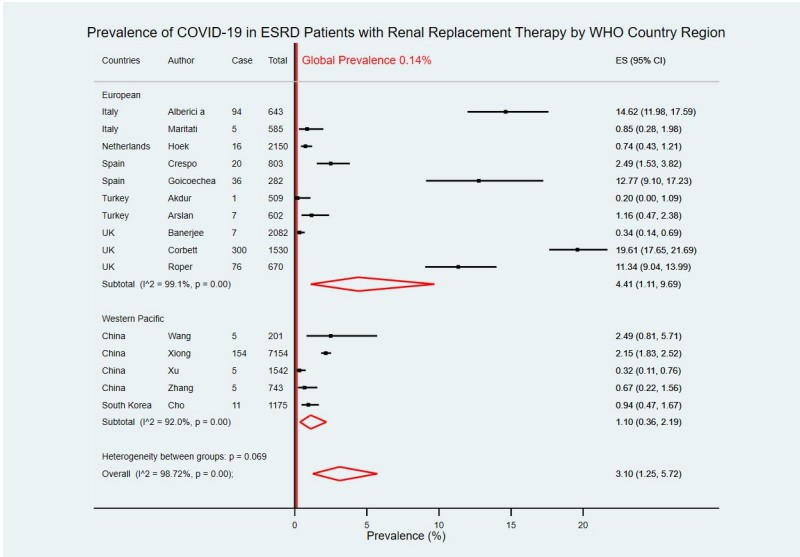

**Fig 4. Forest plot of the prevalence of COVID-19 in ESRD patients with RRT by WHO country region.** The figure summarizes the number of COVID-19 confirmed cases in ESRD patients with RRT and the total number of ESRD patients with RRT in 15 eligible studies with subgroup analysis by WHO country regions. The forest plot represents the estimated prevalence of COVID-19 in ESRD patients with RRT for each study (black boxes), with 95% confidence intervals (95% CI; horizontal black lines). The estimated pooled prevalence for each subgroup was presented with a red diamond. The overall estimated pooled prevalence (last red diamond) was 3.10% (95% CI 1.25–5.72). The global average prevalence (vertical red line) was 0.14%. The meta-analysis used a random-effects model with the exact method for confidence interval estimation. ES, effect size. $I^2$, test for heterogeneity.

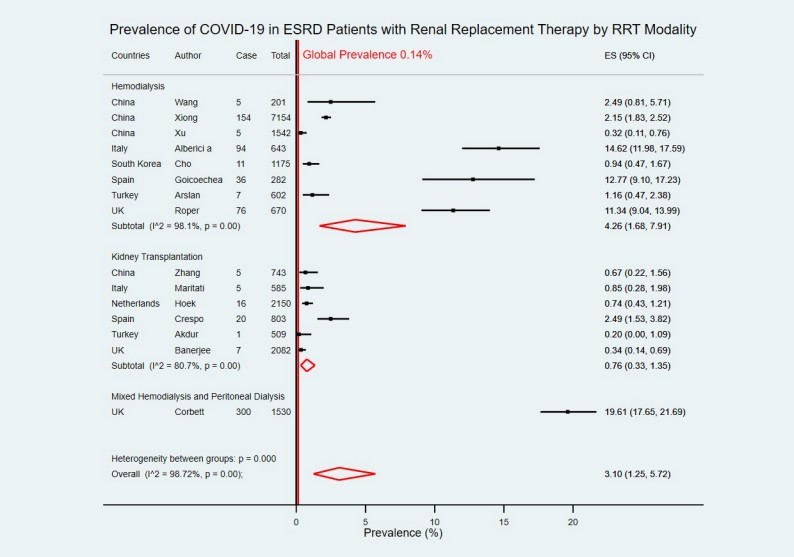

**Fig 5. Forest plot of the prevalence of COVID-19 in ESRD patients with RRT by RRT modality.** The figure summarizes the number of COVID-19 confirmed cases in ESRD patients with RRT and the total number of ESRD patients with RRT in 15 eligible studies with subgroup analysis by types of RRT modality. The forest plot represents the estimated prevalence of COVID-19 in ESRD patients with RRT for each study (black boxes), with 95% confidence intervals (95% CI; horizontal black lines). The estimated pooled prevalence for each subgroup was presented with a red diamond. The overall estimated pooled prevalence (last red diamond) was 3.10% (95% CI 1.25–5.72). The global average prevalence (vertical red line) was 0.14%. The meta-analysis used a random-effects model with the exact method for confidence interval estimation. ES, effect size. $I^2$, test for heterogeneity.

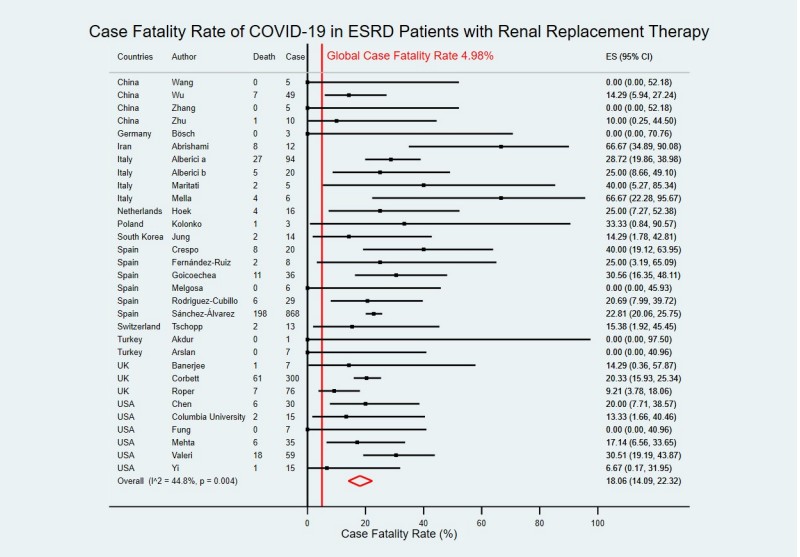

**Fig 6. Forest plot of the case fatality rate of COVID-19 in ESRD patients with RRT.** The figure summarizes the number of COVID-19 deaths in ESRD patients with RRT and the number of COVID-19 confirmed cases in ESRD patients with RRT in 31 eligible studies. The forest plot represents the estimated case fatality rate of COVID-19 in ESRD patients with RRT for each study (black boxes), with 95% confidence intervals (95% CI; horizontal black lines). The estimated pooled case fatality rate (red diamond) was 18.06% (95% CI 14.09–22.32). The global case fatality rate (vertical red line) was 4.98%. The meta-analysis used a random-effects model with the exact method for confidence interval estimation. ES, effect size. $I^2$, test for heterogeneity.

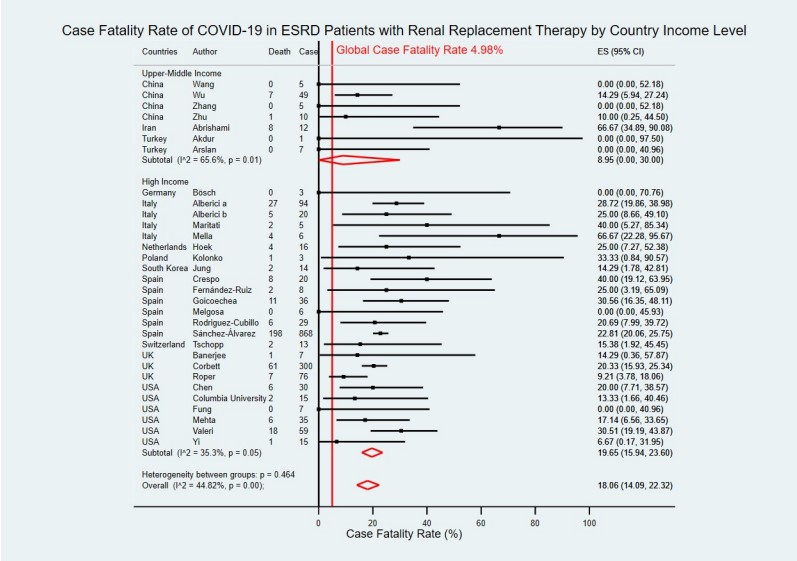

**Fig 7. Forest plot of the case fatality rate of COVID-19 in ESRD patients with RRT by country income level.** The figure summarizes the number of COVID-19 deaths in ESRD patients with RRT and the number of COVID-19 confirmed cases in ESRD patients with RRT in 31 eligible studies with subgroup analysis by the World Bank country income level. The forest plot represents the estimated case fatality rate of COVID-19 in ESRD patients with RRT for each study (black boxes), with 95% confidence intervals (95% CI; horizontal black lines). The estimated pooled case fatality rate for each subgroup was presented with a red diamond. The overall estimated pooled case fatality rate (last red diamond) was 18.06% (95% CI 14.09–22.32). The global case fatality rate (vertical red line) was 4.98%. The meta-analysis used a random-effects model with the exact method for confidence interval estimation. ES, effect size. $I^2$, test for heterogeneity.

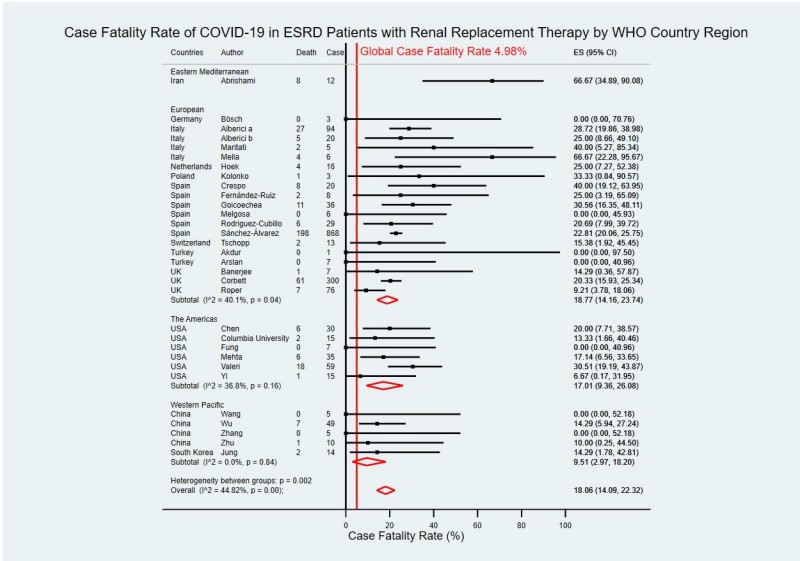

**Fig 8. Forest plot of the case fatality rate of COVID-19 in ESRD patients with RRT by WHO country region.** The figure summarizes the number of COVID-19 deaths in ESRD patients with RRT and the number of COVID-19 confirmed cases in ESRD patients with RRT in 31 eligible studies with subgroup analysis by WHO country regions. The forest plot represents the estimated case fatality rate of COVID-19 in ESRD patients with RRT for each study (black boxes), with 95% confidence intervals (95% CI; horizontal black lines). The estimated pooled case fatality rate for each subgroup was presented with a red diamond. The overall estimated pooled case fatality rate (last red diamond) was 18.06% (95% CI 14.09–22.32). The global case fatality rate (vertical red line) was 4.98%. The meta-analysis used a random-effects model with the exact method for confidence interval estimation. ES, effect size. $I^2$, test for heterogeneity.

European region with an estimated CFR of 18.77% (95% CI 14.16–23.74) [23–25,27–33,36,38–40,42–46], six studies were from the Americas region with an estimated CFR of 17.01% (95% CI 9.36–26.08) [8,47–51], five studies were from Western Pacific region with an estimated CFR of 9.51% (95% CI 2.97–18.20) [19,22,34,35,41], and one study was from Eastern Mediterranean region with CFR of 66.67% (95% CI 34.89–90.08) (Fig 8) [37].

Seven studies focused on COVID-19 CFR in ESRD patients with HD only with an estimated CFR of 14.87% (95% CI 6.87–24.76) [19,23,28,30,33,34,41]. Twenty studies on ESRD patients with KT only reported an estimated CFR of 19.28% (95% CI 12.20–27.20) [22,24,25,27,29,31,35–40,42,44,46–51]. Only four studies had CFR data for mixed RRT modalities (Fig 9) [8,32,43,45].

## Need for mechanical ventilation and intensive care unit (ICU) admission rate

Of 26 studies with hospital admission data, twenty-one studies (80.8%) that reported mechanical ventilation rate of hospitalized COVID-19 cases were included in the meta-analysis of 377 hospitalized COVID-19 cases from nine countries [8,22,24,27,28,31,34–39,41,42,44,46–51]. The overall mechanical ventilation rate of COVID-19 in hospitalized ESRD patients with RRT was 38.75% (95% CI 28.38–49.56).

Of 26 studies reported hospital admission information, fifteen studies (57.7%) from nine countries, which consisted of 223 hospitalized COVID-19 cases in ESRD patients with RRT, reported ICU admission rate [22,24,27,28, 31,34,36–38,41,42,44,46,49,51]. The overall ICU admission rate of COVID-19 in hospitalized ESRD patients with RRT was 28.31% (95% CI 13.84–44.97). Forest plots with subgroup analysis were provided in S2 and S3 Figs.

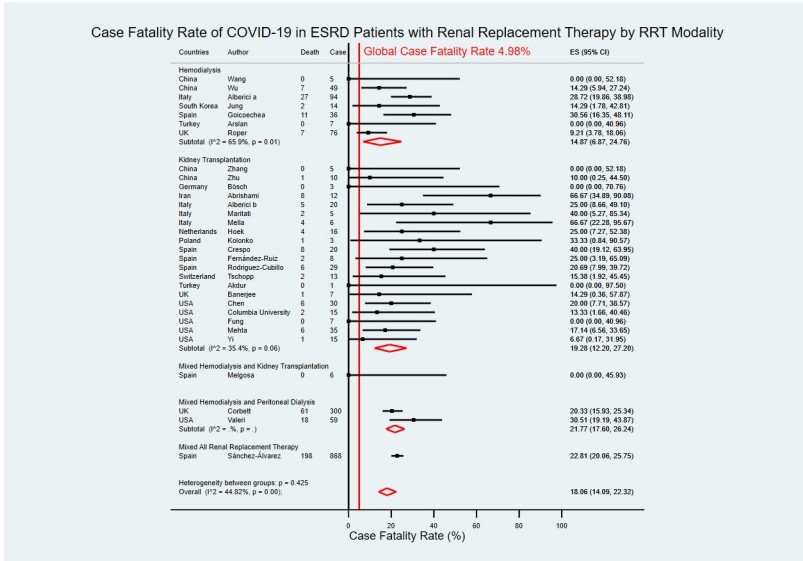

**Fig 9. Forest plot of the case fatality rate of COVID-19 in ESRD patients with RRT by RRT modality.** The figure summarizes the number of COVID-19 deaths in ESRD patients with RRT and the number of COVID-19 confirmed cases in ESRD patients with RRT in 31 eligible studies with subgroup analysis by types of RRT modality. The forest plot represents the estimated case fatality rate of COVID-19 in ESRD patients with RRT for each study (black boxes), with 95% confidence intervals (95% CI; horizontal black lines). The estimated pooled case fatality rate for each subgroup was presented with a red diamond. The overall estimated pooled case fatality rate (last red diamond) was 18.06% (95% CI 14.09–22.32). The global case fatality rate (vertical red line) was 4.98%. The meta-analysis used a random-effects model with the exact method for confidence interval estimation. ES, effect size. $I^2$, test for heterogeneity.

## Discussion

To our knowledge, this is the first systematic review and meta-analysis of studies focusing on the prevalence and CFR of the novel coronavirus in ESRD patients with various renal replacement therapy modalities. The pooled data showed that the prevalence and CFR of COVID-19 in ESRD patients with RRT were significantly higher than the global average in all populations [3]. The need for mechanical ventilation and ICU admission rate in hospitalized ESRD patients with RRT was also high. The outcomes were varied among country income level, WHO region, and type of RRT modality.

The overall pooled prevalence of COVID-19 in ESRD patients with RRT was higher than the global average prevalence (3.1% vs. 0.1%) [3]. Moreover, the overall pooled prevalence of COVID-19 in ESRD patients with RRT was higher than that of autoimmune disease patients (3.10% vs. 1.1%) [53] but was lower than that of colorectal cancer patients (3.1% vs. 45.1%) [54]. The increased prevalence in ESRD patients might relate to decreased immunity from uremia [9].

Despite no evidence of publication bias, there was substantial heterogeneity between studies on prevalence outcomes which might associate with country income level and RRT modality. Significant heterogeneity between upper-middle-income and high-income countries was suggested from the lower estimated pooled prevalence of upper-middle-income countries than high-income countries (0.95% vs. 5.11%). This might be partially explained by the fact that China, which had a relatively lower overall national COVID-19 prevalence than the high-income countries [3], was regarded as an upper-middle-income country. There was also a significant heterogeneity across RRT modalities which the HD patients had a considerably higher prevalence than the KT patients (4.26% vs. 0.76%). The result was unexpected because ESRD patients with KT were supposed to have lower immune function than ESRD patients with HD

due to the use of the immunosuppressive drug [10]. Some ESRD with HD patients might need to visit a hospital for in-center hemodialysis during COVID-19 epidemics and pandemics which might increase the risk of SARS-CoV-2 exposure both from transportation to hospital and visiting hospital [55–57]. A recent meta-analysis on pre-existing hemodialysis patients provided a similar concern with 8.0% COVID-19 incidence in HD patients [58].

The overall CFR of COVID-19 in ESRD patients with RRT was higher than the global average (18.06% vs. 4.98%) [3]. Additionally, the overall estimated CFR of COVID-19 in ESRD patients with RRT was higher than the general population (18.06% vs 0.68 to 5%) [59–61], hypertensive (18.06% vs 6.43%) [62], and autoimmune disease patients (18.06% vs 6.6%) [53]; but was lower than diabetes mellitus (18.06% vs. 19.25%) [63], and cancer patients (18.06% vs. 20.83 to 23.4%) [64,65]. The diminished immune function might play a crucial role in the increased CFR among ESRD patients with RRT [9].

Despite no evidence of publication bias, there was considerable heterogeneity between studies on CFR across WHO regions from 9.51% in Western Pacific to 17.01% in the Americas and 18.77% in the European regions. The variation in CFR might associate with the relatively low national COVID-19 prevalence in China of the Western Pacific region compared to other included countries [3]. There was no difference in CFR among different RRT modalities which might relate to the lack of association of COVID-19 mortality and immunosuppression regimen [66], and surrogates in kidney transplant patients [67]. Subgroup analysis provided a 14.87% case fatality rate in HD patients which was lower than the 25.7% case fatality rate in HD patients from another meta-analysis [58]. The disparity might originate from different eligibility criteria and research objectives between the two studies. That is, while this study aimed to evaluate the prevalence and case fatality rate of COVID-19 in patients with ESRD with RRT, Zhou et al. aimed to evaluate renal involvement and outcomes in hospitalized coronavirus-infected patients.

Several limitations of this systematic review and meta-analysis should be noted. First, the included studies had high heterogeneities for both prevalence and CFR outcomes. The subgroup analysis could not fully explore all sources of heterogeneity, given the limited availability of the information presented in the included studies. Second, there was no included study focused on ESRD patients with PD only. Thus, the prevalence and CFR of COVID-19 in PD patients might not be available. Third, we did not explore deeper into specific details of each RRT modality such as type of HD, type of KT, and duration of RRT. Fourth, all included studies were conducted during the first wave of the COVID-19 pandemic which portrayed an early picture of the COVID-19 global situation. Thus, implications on subsequent COVID-19 waves should be done cautiously as virus variant, vaccination, prevention measures, and treatment might differ. Fifth, all included studies with hospital admission information did not have exclusion criteria of ESRD patients for critical unit or intensive care unit, but only 57.7% (15 of 26) provided sufficient information about intensive care unit stay which might lead to either overestimate or underestimate an actual value. Lastly, we included only English language articles that might miss some pieces of evidence in other languages.

In conclusion, our systematic review and meta-analysis provided the first evidence on the pooled prevalence and CFR of COVID-19 in ESRD patients with RRT which were higher than the global average. Increased prevalence and deaths might relate to reduced immune function in ESRD patients. ESRD patients with RRT should have their specific protocol of COVID-19 prevention and treatment to mitigate excess cases and deaths.

## Supporting information

**S1 PRISMA Checklist.**
(DOCX)

**S1 Fig. Funnel plot: Estimated prevalence and case fatality rate with standard error of each included study in meta-analysis.** (A) estimated prevalence, (B) estimated case fatality rate.
(PDF)

**S2 Fig. Forest plots with subgroup analysis: Mechanical ventilation rate.** (A) overall, (B) by country income level, (C) by WHO country region, (D) by RRT modality.
(PDF)

**S3 Fig. Forest plots with subgroup analysis: ICU admission rate.** (A) overall, (B) by country income level, (C) by WHO country region, (D) by RRT modality.
(PDF)

**S1 Table. Risk of bias assessment.**
(DOCX)

**S2 Table. Full search strategy.**
(DOCX)

## Acknowledgments

The authors thank the investigators of all primary studies that were included in this meta-analysis.

## Author Contributions

**Conceptualization:** Talerngsak Kanjanabuch, Krit Pongpirul.

**Data curation:** Tanawin Nopsopon, Jathurong Kittrakulrat, Kullaya Takkavatakarn, Thanee Eiamsitrakoon.

**Formal analysis:** Tanawin Nopsopon.

**Investigation:** Tanawin Nopsopon, Jathurong Kittrakulrat, Krit Pongpirul.

**Methodology:** Tanawin Nopsopon, Talerngsak Kanjanabuch, Krit Pongpirul.

**Project administration:** Krit Pongpirul.

**Software:** Tanawin Nopsopon, Krit Pongpirul.

**Supervision:** Talerngsak Kanjanabuch, Krit Pongpirul.

**Validation:** Tanawin Nopsopon, Jathurong Kittrakulrat, Krit Pongpirul.

**Visualization:** Tanawin Nopsopon.

**Writing – original draft:** Tanawin Nopsopon, Krit Pongpirul.

**Writing – review & editing:** Tanawin Nopsopon, Jathurong Kittrakulrat, Kullaya Takkavatakarn, Thanee Eiamsitrakoon, Talerngsak Kanjanabuch, Krit Pongpirul.

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
