## [Decision Letter · Decision Letter 0]

14 May 2021

Dear Assoc. Prof. Dr. Pongpirul,

Thank you very much for submitting your manuscript "Covid-19 in end-stage renal disease patients with renal replacement therapies: a systematic review and meta-analysis" for consideration at PLOS Neglected Tropical Diseases. As with all papers reviewed by the journal, your manuscript was reviewed by members of the editorial board and by several independent reviewers. The reviewers appreciated the attention to an important topic. Based on the reviews, we are likely to accept this manuscript for publication, providing that you modify the manuscript according to the review recommendations. 

Sincerely,

Tao Lin, DVM, MSc

Associate Editor

Liesl Zuhlke

Deputy Editor

Reviewer's Responses to Questions

**Key Review Criteria Required for Acceptance?**

**Methods**

-Are the objectives of the study clearly articulated with a clear testable hypothesis stated?

-Is the study design appropriate to address the stated objectives?

-Is the population clearly described and appropriate for the hypothesis being tested?

-Is the sample size sufficient to ensure adequate power to address the hypothesis being tested?

-Were correct statistical analysis used to support conclusions?

-Are there concerns about ethical or regulatory requirements being met?

Reviewer #1: See below

Reviewer #2: METHOD: “We regarded level of heterogeneity for I2 statistic as defined in chapter 9 of the Cochrane Handbook for Systematic Reviews of Interventions: 0–40% might not be important; 30–60% may represent moderate heterogeneity; 50–90% may represent substantial heterogeneity; 75–100% considerable heterogeneity.

There was a problem in the criteria.

Reviewer #3: -Are the objectives of the study clearly articulated with a clear testable hypothesis stated? Yes

 -Is the study design appropriate to address the stated objectives? Yes 

 -Is the population clearly described and appropriate for the hypothesis being tested? YEs

 -Is the sample size sufficient to ensure adequate power to address the hypothesis being tested? Yes

 -Were correct statistical analysis used to support conclusions? Yes

 -Are there concerns about ethical or regulatory requirements being met? No

**Results**

-Does the analysis presented match the analysis plan?

-Are the results clearly and completely presented?

-Are the figures (Tables, Images) of sufficient quality for clarity?

Reviewer #1: See below

Reviewer #2: Yes

Reviewer #3: -Does the analysis presented match the analysis plan? Yes. Some additional information about authors response rates should be added

 -Are the results clearly and completely presented? Yes

 -Are the figures (Tables, Images) of sufficient quality for clarity Yes

**Conclusions**

-Are the conclusions supported by the data presented?

-Are the limitations of analysis clearly described?

-Do the authors discuss how these data can be helpful to advance our understanding of the topic under study?

-Is public health relevance addressed?

Reviewer #1: See below

Reviewer #2: Yes

Reviewer #3: This is an intersting, well designed and well conducted metanalysis about risk of COVID in ESRD/RRT.

Authors conclude in line with a majority of studies that patients under RRT are in higher risk for severe COVID and mortality rates as high as 18%. I consider it as a very valuable paper and probably will be highly referenced since it resume current evidence in risk of mortality for ESRD.

Corrections that should be done and suggestions:

Since the aim of study is to include different regions of the world, I suggest to include an EU reference with more that 1600 COVID cases, similar to REF 5 and to support comments about AKI-COVID in page 4 line 87 REF 12-14 (Nephrology Dialysis Transplantation, Volume 35, Issue 8, August 2020, Pages 1353–1361, https://doi.org/10.1093/ndt/gfaa189) 

Methods (details concerning author consultation) seems to be duplicate in pag 5 line 121 and page 6 line 137. Correct it please

It would be usefull to give informmation about non-responding authors rates.

Page 11 line 213 typo error, space after brakets

Some of the selected papers include data regarding the policy of exclussion of ESRD patient for Critical Unit. This point is highly relevant and shoul be added and commented (i.e. after data in page 6 line 332), A majority of studies came from the 1st pandemic wave. Some coment could be done regarding potential differences with susequent waves and the external validity of theese results.

**Editorial and Data Presentation Modifications?**

Reviewer #1: (No Response)

Reviewer #2: Minor Revision

Reviewer #3: See above

**Summary and General Comments**

Reviewer #1: Nopsopon and colleagues performed a systematic review and meta-analysis on mortality in patients with end-stage kidney disease and maintenance dialysis. This is a timely and important work. My comments are provided below:

Major comments:

- Why were patients with kidney transplantation considered as ESRD patients? I would assume that they are a separate group. The mechanisms of COVID-19 disease severity are likely to be different from uremia in patients with kidney transplantation. Unless the authors don't have a valid reason, I would recommend to exclude the group

- Why did the authors only include studies up to June 2020 (i.e. 1 year before the submission of this work)?

Minor comments:

- The manuscript requires serious proofreading to improve the readability

- Figure legends are normally placed at the end of the manuscript

Reviewer #2: This is a well-performed systematic review and meta-analysis about infection and mortality rates of the SARS-CoV-2 infection in ESRD patients who are on RRT. My minor comments are listed below.

1 ABSTRACT and DISCUSSION: “There was no systematic study on the infection and mortality of the SARS-CoV-2 infection in ESRD patients who are on RRT.”

A recent meta-analysis and systematic review found that the incidence of coronavirus infection was 7.7% (95% CI: 4.9%-11.1%) in prevalent hemodialysis patients with an overall mortality rate of 26.2% (95% CI: 20.6%-32.6%) (Ren Fail. 2020 Nov 9;43(1):1-15). The difference between this article and the review by Dr. Zhou et al. should be discussed in the discussion part.

2 line 352 ‘KTwere’ , line 363 ‘CFRacross’.

3 Subgroup analysis could add one according to the tropical and frigid zone if possible.

Reviewer #3: This is an intersting, well designed and well conducted metanalysis about risk of COVID in ESRD/RRT.

Authors conclude in line with a majority of studies that patients under RRT are in higher risk for severe COVID and mortality rates as high as 18%. I consider it as a very valuable paper and probably will be highly referenced since it resume current evidence in risk of mortality for ESRD.

Minor Corrections that should be done and suggestions are provided

PLOS authors have the option to publish the peer review history of their article (what does this mean?). If published, this will include your full peer review and any attached files.

Reviewer #1: No

Reviewer #2: Yes: Cheng Xue

Reviewer #3: No

Figure Files:

Data Requirements:

Reproducibility:

References

---

## [Editor Report · Decision Letter 1]

28 May 2021

Dear Assoc. Prof. Dr. Pongpirul,

We are pleased to inform you that your manuscript 'Covid-19 in end-stage renal disease patients with renal replacement therapies: a systematic review and meta-analysis' has been provisionally accepted for publication in PLOS Neglected Tropical Diseases.

Best regards,

Tao Lin, DVM, MSc

Associate Editor

Liesl Zuhlke

Deputy Editor

---

## [Editor Report · Acceptance letter]

10 Jun 2021

Dear Assoc. Prof. Dr. Pongpirul,

We are delighted to inform you that your manuscript, "Covid-19 in end-stage renal disease patients with renal replacement therapies: a systematic review and meta-analysis," has been formally accepted for publication in PLOS Neglected Tropical Diseases.

Best regards,

Shaden Kamhawi

co-Editor-in-Chief

Paul Brindley

co-Editor-in-Chief
